# Alcohol unleashes homo economicus by inhibiting cooperation

**Paul J. Zak** [1]*, **Kylene Hayes**[1], **Elizabeth Paulson**[1], **Edward Stringham**[2]

**1** Center for Neuroeconomics Studies, Claremont Graduate University, Claremont, CA, United States of America, **2** Department of Economics, Trinity College, Hartford, CT, United States of America

* paul.zak@cgu.edu

## Abstract

Human behavior lies somewhere between purely self-interested *homo economicus* and socially-motivated *homo reciprocans*. The factors that cause people to choose self-interest over costly cooperation can provide insights into human nature and are essential when designing institutions and policies that are meant to influence behavior. Alcohol consumption can shed light on the inflection point between selfish and selfless because it is commonly consumed and has global effects on the brain. The present study administered alcohol or placebo (N = 128), titrated to sex and weight, to examine its effect on cooperation in a standard task in experimental economics, the public goods game (PGG). Alcohol, compared to placebo, doubled the number of free-riders who contributed nothing to the public good and reduced average PGG contributions by 32% (p = .005). This generated 64% higher average profits in the PGG for those who consumed alcohol. The degree of intoxication, measured by blood alcohol concentration, linearly reduced PGG contributions (r = -0.18, p = .05). The reduction in cooperation was traced to a deterioration in mood and an increase in physiologic stress as measured by adrenocorticotropic hormone. Our findings indicate that moderate alcohol consumption inhibits the motivation to cooperate and that *homo economicus* is stressed and unhappy.

## Introduction

Cooperation in one-shot settings is ubiquitous. But then so are defection and free-riding. A substantial set of mathematical models, laboratory experiments, and empirical analyses have sought to determine when people are likely to cooperate or be selfish [1–4]. Cooperative behaviors are predicted by multi-level evolutionary models in which individuals in a group out-compete other groups [5,6]. Costly signaling of one's value as a future collaborator, known as indirect reciprocity, also supports cooperation [7–10]. But, indirect reciprocity requires observation of behavior by others. A study with 136,000 private contributions to Social Funds by students at the University of Zurich showed that two-thirds donated money [11]. Similarly, 1,500 randomly selected residents of Denmark reported that 69% were conditional cooperators and 15% were free riders [12]. The conditions under which people cooperate or defect are still not fully understood.

**Data Availability Statement:** The data are available at Open ICPSR https://doi.org/10.3886/E136561V1.

**Funding:** ES provided funding from the Davis Endowment at Trinity College.

**Competing interests:** The authors have declared
that no competing interests exist.

Cooperation been studied in the laboratory using a variety of social dilemmas [13,14]. One
used extensively is the public goods game (PGG) [15]. In this task, groups of participants can
contribute part of their endowments for a social goal. The dilemma arises because each mem-
ber equally shares in the aggregate contribution, providing an incentive to free-ride. Neverthe-
less, contributions are made in many variants of the PGG [16–21]. A meta-analysis of 27 PGG
studies reported an average contribution of 38% of participants' endowments [22]. Typically,
one-third to one-half of participants in PGG experiments are free-riders [23,24]. Contribu-
tions are moderate in single-decision games, decline with repeat play, but increase when deci-
sions are made face to face [25]. Possible explanations include preferences for altruism and
reciprocity [26–29], confusion [30–32], and risk of punishment [18].

The present paper seeks to understand the inflection from cooperation and defection by
investigating the impact of an activity that more than one-half of Americans [33] and even
more Europeans do at least once a month [34]: consume alcohol. Alcohol has global effects on
the brain and impacts decision-making [35]. The primary neurochemical effect of alcohol is
increased activity of the inhibitory neurotransmitter GABA [36]. GABA is responsible, in part,
for the disinhibition and impulsivity typically observed in drinkers [37]. A secondary effect of
alcohol is to stimulate the activity of the excitatory neurotransmitter dopamine in the ventral
striatum [38]. Striatal activity is associated with reward-seeking behaviors [39,40]. These neu-
rochemical factors are part of the reason for alcohol motivates a desire for immediate rewards
[41,42]. The father of scientific psychology, William James called wrote that "Alcohol [is]. . .the
great exciter of the Yes function in man" [43].

Despite an extensive literature on alcohol and impaired decision-making, little is known
about how acute, rather than long-term, alcohol use affects cooperative behaviors. Alcohol use
has been reported to weakly reduce charitable donations [42]. Yet, alcohol imbibers were
equally generous in a strategic-share-the money laboratory task but and the same time rejected
unfair offers more often than sober participants [44]. This behavior may be due to alcohol con-
sumption inhibiting the ability to recognize emotions in others and diminishing empathy
[45,46]. At the same time, alcohol use inhibits individuals' abilities to emotionally regulate
themselves [47] that can manifest as selfishness. Yet, male drinkers, but not females, promised
to cooperate in the prisoner's dilemma game at a higher rate than did their teetotaler brethren
[48].

Outside of laboratory tasks, there is evidence that casual alcohol consumption affects social-
economic outcomes. Analysis of General Social Survey showed that drinkers earn 10–14%
more than abstainers and that those who drink socially earn an additional 7% [49]. These
authors posit that moderate drinkers spend more time socializing with colleagues and thereby
build social capital, although the literature is unclear on the alcohol to social capital association
[50]. In order to untangle this relationship, a controlled laboratory study was run to assess the
behavioral effects of acute alcohol consumption on group cooperation as well as neurobiologi-
cal and psychological mechanisms driving behavior.

## Methods

The Institutional Review Board of Claremont Graduate University approved this study (IRB
#2175) with sessions held at the Center for Neuroeconomics Studies in Claremont, CA. Writ-
ten informed consent was obtained from all participants before they were included in the
experiment. There was no deception of any kind and participants were assigned an alphanu-
meric code to mask their identities. Participants were informed that that they would either
consume alcohol or a placebo during the study. Tasks were incentivized with money and
participants could earn up to $65 depending on their decisions and decisions of others.

**Fig 1. Timeline of the experiment.**

Anonymity was maintained by having a lab administrator who was not associated with the study pay participants their earnings in private at the study's conclusion.

## Study timeline

Fig 1 shows time course of the study. After consent, participants completed surveys assessing their opinions, attitudes, and demographics and were weighed. Participants were then led to a private room where their basal blood alcohol concentration (BAC) was established using a commercial breathalyzer (BACtrack S35, San Francisco, California) and a blood draw was done to establish basal endocrine levels. Participants were excused if their BAC > 0.001%. A die roll next assigned participants to consume drinks alone or in groups of four. Drinks either contained alcohol or a similar-tasting placebo. Participants then had a second blood draw after which they made a social decision and completed a final survey.

## Alcohol administration

The protocol sought to induce a belief by participants' that they were consuming alcohol to mask the placebo even though the consent form stated that only one-half of participants would receive alcohol. This was done in several steps following previous protocols used in previous studies involving alcohol and social interactions [51,52]. This including have a bottle marked "vodka" on a drink cart where a research assistant mixed 1.5–3.0 ounces of 80-proof SKYY vodka (Groupo Campari, Milan Italy) or an identical tasting placebo vodka (Arkay Alcohol Free Vodka, Jalisco, Mexico) with cranberry juice in clear highball glasses. Each drink consisted of one-part vodka or placebo, 3.5 parts cranberry juice. Males (females) received 62 (56) grams vodka per kilogram of body weight [53]. The expected BAC levels were 0.04–0.06%. Previous studies involving social alcohol consumption administered males (females) with 82 (74) grams vodka per kilogram body weight, resulting in BAC levels of 0.6–0.8% [54]. We reduced the amount administered to minimize the risk of any participant nearing the California legal driving limit of 0.08%. Participants were randomly assigned to the alcohol (A) or placebo (P) conditions. The rims of all glasses were dabbed with vodka in advance to induce a stinging sensation on the tongue to further influence P participants' beliefs they were consuming alcohol. These procedures have led participants in the placebo- condition to believe they had consumed alcohol in previous studies [55,56]. Alcohol metabolism varies with a person's weight and sex [57]. The drinks were mixed by the experimenters in front of participants and they were served one third of their drink every 10 minutes, over the course of 30 minutes. Participants had their BAC measured before being served subsequent portions of their drink. BAC levels were not shown to participants. After the conclusion of data collection, participants were not dismissed from the lab for safety reasons until their BAC reading was less than 0.03%.

## Participants

All participants who were of legal drinking age (21) and identified themselves as social drinkers. Candidate participants were excluded if they were i) active alcoholics or at high risk of

alcohol abuse following National Institute on Alcohol Abuse and Alcoholism standards [58]; ii) Asians and Native Americans due to low penetrance of alcohol dehydrogenases genes; iii) pregnancy and medical conditions contraindicating alcohol consumption; iv) being 15% above or below ideal height/weight ratio (ideal range 18.50–24.99 kg/m$^2$) [59] that makes dosing alcohol problematic.

Participants were randomly assigned to consume drinks for 30 minutes during which they were served 3 drinks at 10 minute intervals. BAC was measured before the second and third drink were consumed and 10 minutes after the third drink. Participants' BACs were monitored during drinking to ensure consumption occurred and the desired BAC range was achieved.

Participants were randomly assigned to drink alone or in a group with three other participants who were strangers [52]. During intake, participants were asked to identify if they knew other participants to ensure all groups contained strangers. All groups had a mix of sexes to capture the atmosphere in a typical bar setting. The assignment to drink alone or in a group was included to isolate the effect of socializing with others from the effect of alcohol. While consuming their drinks in groups, participants were instructed to discuss whatever they like except for the subsequent portion of the study, past study participation, and their level of intoxication. Using the average size effect and standard errors in Barraza & Zak [60] produces a power of test of 0.99 using G$^*$power [61] for 100 observations.

## Behavioral task

Participants were instructed to move to a bank of computers with partitions to make a decision involving money. A task known as the Public Goods Game (PGG) [20,25] asks participants to contribute to a common pool that is then shared by all as a measure prosocial behaviors. All participants were equally and fully instructed in the task, including being shown examples, and were given a chance to ask the experimenters questions. The PGG was presented using zTree software that uses standard instructions that avoid priming individuals to contribute money [62]. The instruction used neutral language throughout that avoided identifying other participants as "partners" or "competitors" that has been shown to influence decisions when sharing money with others [63]. Participants were endowed with $10 and could contribute any sum, including zero, to a common pool of three other unknown participants in the experimental session. Decisions were made in private in partitioned computer stations and no discussion among group members was allowed. Typical sessions included 12 participants and zTree software randomly formed groups of four in the PGG so participants would not know if their PGG group include others with whom they had socialized. Sessions that had less than 12 participants (4) used a research assistant in the control booth to complete the group of four so the session could proceed. The research assistant was not the one who mixed drinks, nor did he or she drink, and was instructed to make the median PGG contribution from the literature. These four fill-in observations were not included in the analysis and participants did not see others' contributions so this approach could not affect participants' decisions. Participants were randomly assigned to groups of four in order to reduce any pre-play social attachment. Participants were not informed about the PGG while drinking in order to reduce anticipatory effects. The instructions stated they would "make decisions" after consuming drinks.

The instructions stated that the total amount contributed to the common pool would be tripled and then split evenly among the four contributors. Those who contribute benefit others in the group, but risk being taken advantage of by non-contributors ("free-riders"). Earnings are highest when a participant free-rides by contributing zero while others contribute nonzero amounts [64]. Participants made four decisions in the PGG without feedback and were

informed that zTree would randomly re-match them with others in each round. This was done to inhibit the possible effect of reputation on decisions [65]. Contributions to the common pool and earnings were averaged as dependent variables in order to reduce confusion during decisions in a novel task [19,66].

## Blood draws and assays

Participants had two blood draws from an antecubital vein by a qualified phlebotomist. Two 8-ml EDTA whole-blood tubes were drawn in a sterile field using Vacutainer® blood-collection kits. Tubes were stored on ice before being placed in a refrigerated centrifuge and spun at 1500 rpm at 4˚C for 12 min following previous protocols [67]. Plasma was aliquoted into 2-ml polypropylene Fisher brand microtubes that were immediately placed on dry ice and then transferred to an -80˚C freezer until analysis. Adrenocorticotropic hormone (ACTH) is a fast-acting hormone associated with arousal and stress [68] and was assayed to determine how moderate alcohol intact affects physiology. ACTH was assayed using radioimmunoassay (RIA) produced by DiaSorin, Inc. (Stillwater, MN, USA) by the Reproductive Endocrine Research Laboratory at the University of Southern California (USC). The inter-assay CVs < 11% for all analytes.

## Surveys

Questionnaires were used to measure demographics, one's perceived intoxication (SIS: Subjective Intoxication Scale) [51] acute changes in mood (PANAS: Positive Affect Negative Affect Scale) [69], Satisfaction with Life survey (SWL) [70], and closeness to others (IOS: Inclusion of Others in Self) [71].

## Statistical analysis

Student's t-tests (denoted "t") were used to examine the hypothesized relationship between prosocial behaviors and alcohol consumption by comparing means and testing of Pearson's correlations are nonzero. Several samples had unequal variances assessed using Levene's test and for these Welch's t-test was used and noted in the analysis. The impact of affect and isolation release was examined using a mediation model. We used two key dependent variables to demonstrate the robustness of our findings, contributions in the PPG and profits earned from the PGG.

# Results

A total of 128 individuals participated in the experiment (male: n = 58; female: n = 70). Participants did not vary pre-treatment in SWL, IOS, PANAS, basal ACTH, age, or sex distribution (ps>.20). No gender differences in basal states were found (ps>.05).

## Behavior

The treatment effectively raised mean BAC compared to placebo (A: M = 0.05, SD = 0.03; P: M = 0.00, SD = 0.00; N = 124, Welch's t = 15.22, p = .0001, d = 2.82). Participants in the treatment group reported feeling intoxicated (A: M = 19.67, *SD* = 18.85; P: M = 6.38, *SD* = 9.58; N = 124, Welch's *t*-test = 5.03, *p* < .001, d = 0.89) with BAC and SIS positively correlated (r = 0.40, t = 4.766, p = .0001).

The contributions to the common pool in the PGG (PGGc) were lower in the treatment group compared to placebo (A: M = 3.61, SD = 3.25; P: M = 5.32, SD = 3.55; t = 2.83, p = 0.005, d = 0.50). The reduction in contributions induced by alcohol in the mixed treatment and placebo sessions resulted in higher profits (PGGp) for treatment participants (A:

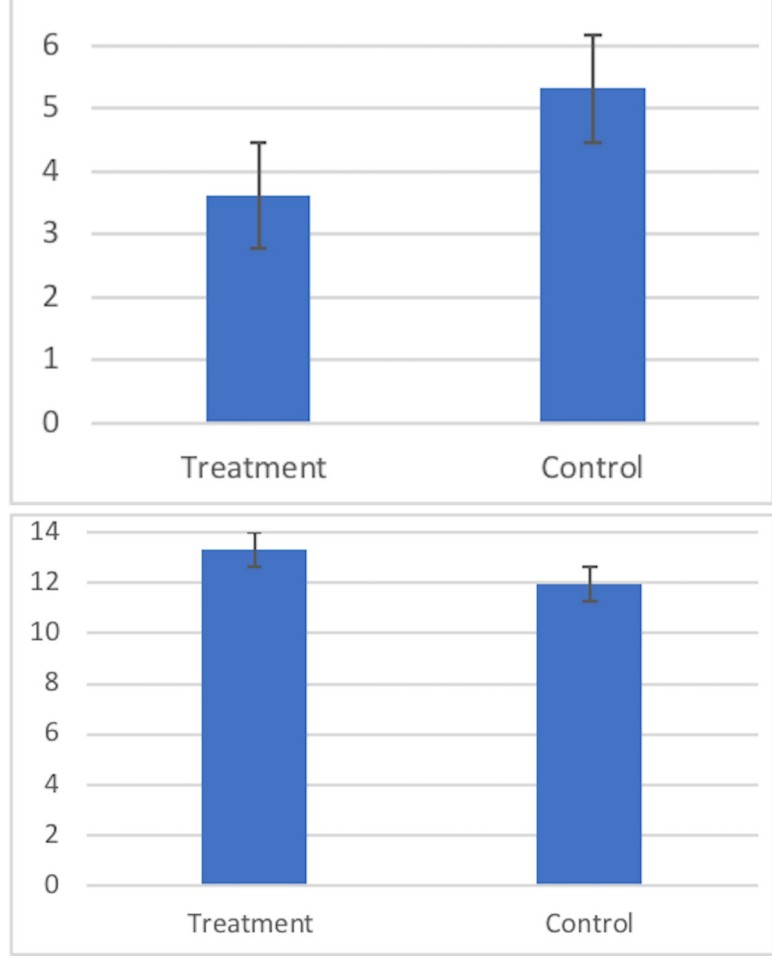

**Fig 2.** A. Alcohol reduced PGG contributions by 32% (p = .005) and B. increased earnings from the PGG by 11.5% (p = 0.006) showing that alcohol increased free-riding. Bars indicate SEs.

M = 13.33, SD = 2.95; P: M = 11.96, SD = 2.59; t = 2.79, p = 0.006, d = 0.49; Fig 2). There were no gender differences and no effects of drinking in isolation on PPGc or PPPp (ps> .40).

In order to confirm that alcohol consumption reduced PGGc, a panel data linear regression was estimated (n = 496) in which each previous round's contribution was included as an independent variable along with indicators for alcohol and socializing. The alcohol treatment continued to significantly reduce PGG contributions (B = -0.57, one-tailed t-test p = .033) and the previous contribution was also significant (B = 0.59, one-tailed t-test p < .0001). The regression did not suffer from multicollinearity (VIFs<1.10). Two more analyses were done to check the robustness of the results. First, PGG round one contributions were assessed using the same variables except the previous round contribution. Alcohol consumption again significantly reduced PGGc (B = -1.79, one-tailed t-test p = .005). Second, the analysis we repeated for round four contributions. Alcohol again reduced PGGc (B = -1,46, one-tailed t-test p = .017). The p-values are nearly identical in all regressions when age and sex are included as controls.

Next, we assessed other behavioral measures of cooperation. The effect of free-riding by treatment can be seen as the return on their PGG investment (PPGp/PGGc) for the 100 participants who contributed a nonzero amount to the common pool. Those who consumed alcohol had a 369% return on their investment compared to a return of 225% return by placebo

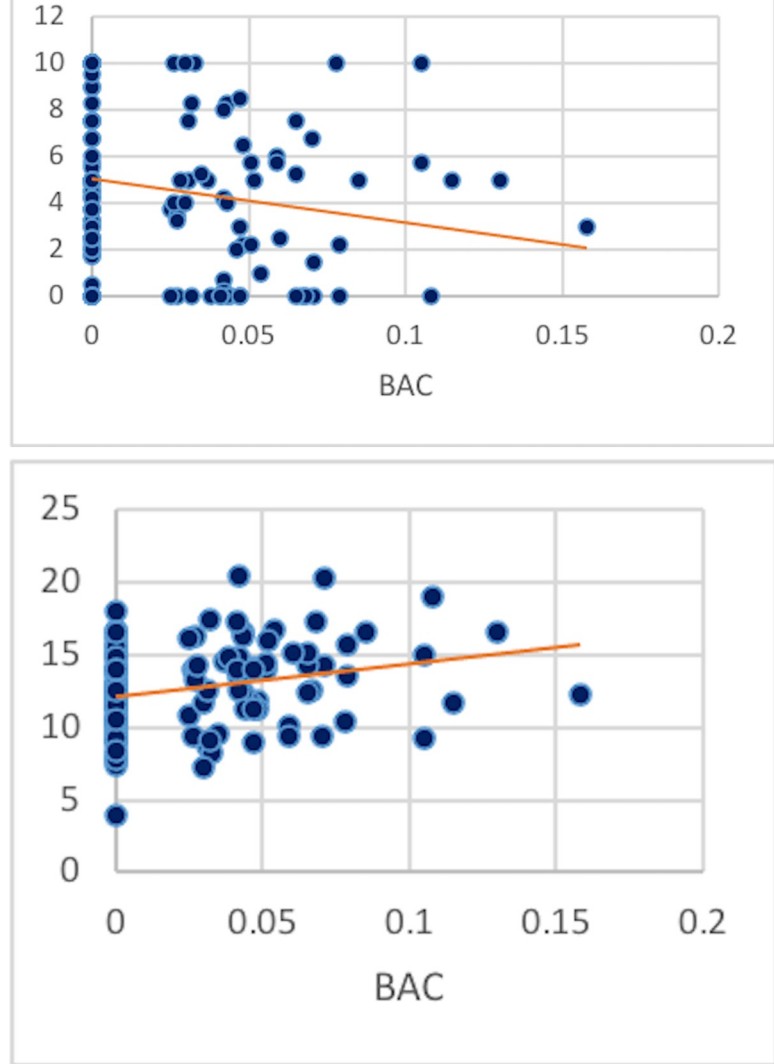

**Fig 3.** A. PGG contributions were inversely associated with BAC levels (r = -0.18) while B. PGG earnings increased linearly with BAC (r = 0.27).

participants. Alcohol produced twice as many complete free-riders, i.e. those who contributed nothing (A: n = 19; P: n = 9). We estimated a logistic regression to assess the accuracy in predicting free-riders. BAC, sex, age, and post-drinking ACTH were included as independent variables. The model correctly classified free-riders with 80.7% accuracy (pseudo-$R^2$ = .14. p = .003) and absent multicollinearity (VIFs $\leq$ 1.03).

The amount of intoxication (BAC) directly reduced PGG contributions (r = -0.18, t = 2.00, p = .05) and increased participant profits (r = 0.27, t = 3.06, p = .003, Fig 3). The association between BAC and PGG contributions and profits were confirmed by the impact of participants' self-reported intoxication (SIS) on PGG contributions (r = -.20, t = 2.29, p = .02).

## Stress and affect

Both control and treatment participants had an increase in negative affect during the study (change in negative affect: -0.171, t = -11.875, df = 127, *p* = .0000). This result continued to

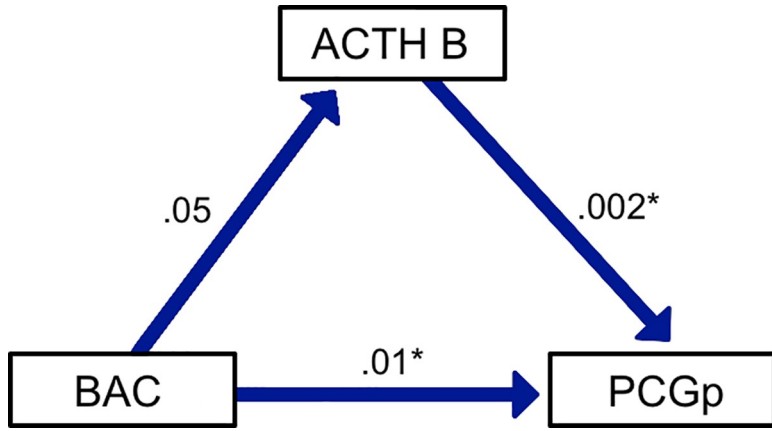

**Fig 4. A mediation model shows that alcohol (BAC) directly increases profits in the PGG (p = .013) and trends toward an indirect impact by increasing stress (ACTH, p < .053).** Standardized regression coefficients are shown, * = p < .05.

hold when controlling for age and sex (ps < .05). Whether participants drank alone or in a group did not affect the change in negative affect (p = .18) nor did participant sex (p = .23). The data showed that alcohol intake increased physiologic stress. Post-treatment ACTH trended towards parametric relationship with BAC (r = 0.18, t = 1.92, *p* = 0.058). This relationship became significant when controlling for age and sex in a linear regression (BAC: B = -.928, p = .037). The stress effect was driven both by alcohol and by isolation (B = .533, p = .034).

We estimated a model for PGGp to test if stress (ACTH) mediates the increase in profits in the PGG. The model shows that alcohol increases stress directly and both directly and indirectly increases profits in the PGG (p = .003). A mediation model for PGG contributions had similar estimates (p = .043; Fig 4).

## Discussion

We found that moderate alcohol consumption reduced contributions to a public good pool by 32%. Those who consumed alcohol earned 64% more money because they interacted with more cooperative placebo participants. Alcohol also doubled the number of participants who were complete free-riders, contributing nothing to the public good. BAC linearly reduced PGG contributions "unleashing" individuals to behave selfishly.

If money is the only value one receives from cooperation, at least as captured by the PGG, then the present study has shown that a moderate consumption of alcohol results in behavior closer to that predicted by traditional models in economics [2,18,72]. This may be due to alcohol's stimulation of the neurotransmitter dopamine [38] that is strongly associated with reward-seeking behaviors [73]. Conversely, a rich literature has documented the humans are gregariously social and that most people in most circumstances are biased towards cooperation [74,75]. Our results are unlikely to be affected by the methodology we employed. Previous research has shown that monetary decisions that capture cooperative behaviors that include blood draws match those absent blood sampling [74]. Nevertheless, while we sought to capture typical social drinking, our results may not generalize to single-sex alcohol consumption or drinking by older cohorts.

Alcohol's inhibition of appropriate social responses has been termed "alcohol myopia" [76] but is more typically seen in heavy drinkers and alcoholics that moderate imbibers [77]. The primary mechanism producing inhibition has been traced to an increase in the inhibitory

neurotransmitter GABA [37]. If the treatment reported here resulted in alcohol myopia, it appeared to decrease the value put on social benefits and increase the value of selfish benefits as has been shown with testosterone administration [78]. Indeed, pre-play communication has been consistently been shown to increase cooperation [79], yet alcohol was shown to blunt this effect.

Alcohol's reduction of the perceived value of cooperation was manifested in the present study by an increase in negative affect. Alcohol accentuates emotional volatility [76,80], negative affect [81], and impulsivity [82]. The reduction in prefrontal activity that moderates social-emotional responses [83] reduced affective states in alcohol-consuming participants and may have focused them on immediate monetary rewards rather than the psychic reward of conforming to a social norm of cooperation [84]. Most economics studies have measured impulsivity by the choice of immediate versus delayed rewards. Individuals showing patience for temporal rewards are generally more cooperative [85] counter to the results found here. The role of stress has not been measured in the existing literature and may explain the difference in findings. At the same time, trait impulsivity can lead to alcohol use and abuse [86].

Social rejection and physical pain have been also shown to increase one's desire for money [87] and our analysis suggests that an increase in negative affect of moderate alcohol consumption may mimic pain responses when it comes to money [88]. This finding is in contrast to much of the literature showing that negative affect increases monetary allocations to others in ultimatum and dictator games [89]. We did not find that isolation while drinking influenced negative affect compared to those drinking socially as others have reported [90].

Perhaps our most valuable finding is that alcohol increases physiologic stress and through this route reduces cooperation. This was captured by higher levels of the stress hormone ACTH for those who consumed alcohol. ACTH, rather than cortisol, was measured because the former responds more quickly than the latter in line with the time course of the experiment. Moderate stress tends to increase prosociality [91–93] while high and/or chronic stress inhibits prosocial behaviors [94,95]. Moderate alcohol consumption may be an effective way to induce physiologic stress, in particular, by having people drink alone. Our finding that physiologic stress was higher for those drinking alone seems to be new in the literature that has focused on drinking to reduce stress [96]. While drinking alone is a known risk factor for alcohol abuse [90], we have shown that drinking alone reduces subsequent prosocial behaviors. This may further isolate and stress solo drinkers, influencing them to continue to imbibe alcohol. The increase in stress was primarily driven by women and we believe this finding is worth additional research. A replication of study's results is warranted because when segmented into subsamples, some of the analysis was relatively underpowered.

Our findings show that *homo economicus* is alive and well and that alcohol is enough to bring him out. A variety of factors besides alcohol reduce prosocial tendencies, including high levels of testosterone and serotonin depletion [97]. The present study was not designed to capture the contribution of changes in neurotransmitters on cooperation, but this is a rich area for future research.

## Acknowledgments

We thank Mr. Garrett Thoelen who organized and supervised the data collection and database construction. We also recognize the editor and two reviewers for excellent comments that have improved this manuscript.

## Author Contributions

**Conceptualization:** Paul J. Zak, Edward Stringham.

**Formal analysis:** Paul J. Zak, Kylene Hayes, Elizabeth Paulson.

**Funding acquisition:** Edward Stringham.

**Investigation:** Edward Stringham.

**Methodology:** Paul J. Zak, Edward Stringham.

**Project administration:** Paul J. Zak.

**Resources:** Paul J. Zak.

**Software:** Paul J. Zak.

**Supervision:** Paul J. Zak.

**Validation:** Paul J. Zak.

**Visualization:** Elizabeth Paulson.

**Writing – original draft:** Paul J. Zak, Kylene Hayes.

**Writing – review & editing:** Paul J. Zak, Edward Stringham.

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
