## [Decision Letter · Decision Letter 0]

23 Mar 2021

PONE-D-21-04756

Alcohol Unleashes Homo Economicus by Inhibiting Cooperation

PLOS ONE

Dear Dr. Zak,

Thank you for submitting your manuscript to PLOS ONE. After careful consideration, we feel that it has merit but does not fully meet PLOS ONE’s publication criteria as it currently stands. Therefore, we invite you to submit a revised version of the manuscript that addresses the points raised during the review process.

You will find both reports pasted below. I agree with them that certain details should be clear. Regarding power calculations I am not asking for new experimental sessions. It would be enough if you just recognize that certain results might be taken cautiously due to the lack of power (since you have many treatment and quite limited sample). 

We look forward to receiving your revised manuscript.

Kind regards,

Pablo Brañas-Garza, PhD Economics

Academic Editor

PLOS ONE

Journal Requirements:

'no'

a. Please complete your Competing Interests statement to state any Competing Interests. If you have no competing interests, please state "The authors have declared that no competing interests exist.", as detailed online in our guide for authors at http://journals.plos.org/plosone/s/submit-now

4. Please ensure that you refer to Figure 4 in your text as, if accepted, production will need this reference to link the reader to the figure.

Reviewers' comments:

Reviewer's Responses to Questions

**Comments to the Author**

1. Is the manuscript technically sound, and do the data support the conclusions?

Reviewer #1: Yes

Reviewer #2: Yes

2. Has the statistical analysis been performed appropriately and rigorously? 

Reviewer #1: Yes

Reviewer #2: Yes

3. Have the authors made all data underlying the findings in their manuscript fully available?

Reviewer #1: Yes

Reviewer #2: Yes

4. Is the manuscript presented in an intelligible fashion and written in standard English?

Reviewer #1: Yes

Reviewer #2: Yes

5. Review Comments to the Author

Reviewer #1: The paper is well done and the conclusions are interesting. I have only two small comments. One is that I do not know why the authors did the two treatments where participants drank alone or in a group. This ends up not making a difference, but I suppose they conjectured it could have been different and I would like to understand this.

The other point is that the closest study I can see is the one of of Corazzini, Filippin & Vanin, 2015. I am surprised that they chraracterize it as "Alcohol use has been reported to weakly reduce charitable donations." I was curious and downloaded the paper. It seems like the donations to the both the charity MSF and the journal are reduced by about a half of the baseline. So in a sense it is very much in line with what the authors find here, if anything a bit higher. One added value of this study in my view is to do it in a strategic setting, on the one hand. Did the authors think the strategic setting would change something? If so, why? If not, what have we learnt? The other added value is to check the mediation of stress, which the other study did not check. I have a sense this is the major contribution, and perhaps the authors could highlight it a bit more.

Reviewer #2: I have very much interest in this paper. However, at this point I have several important concerns, which I shorlty list here:

1. The sample is clearly too small. 128 individuals, split into 2 treatements (control and intervention), plus in 2 again (drink alone or drink with 3 people). I would expect more sessions to be conducted.

2. I would need more details about the sessions. It is said that a session consists of 12 people, how were exacly composed the sessions since with 128 people you cannot have only sessions of 12?

3. Why were there mixed groups? There is a large litterature showing that men and women react differently to alcohol and behavioural tasks, why not making groups of females only and of men only?

4. It is said there was no deception, however some participants had placebo and it is said they were pushed to believe they will have alcohol. So I am sorry, but I cannot understand how this was not deception?

5. It is not clear if participants knew the task in advance, while drinking, and if they knew there will be a task at all?

6. It is explained that alcohol makes people act on impulse, but economic papers show that when acting on impulse, people are more cooperative. So I would appreciate a discussion.

7. It is said that alcohol motivates the desire for immediate reward, but this reward could be warm glow or something like this, so, again, I would need a discussion.

8. There is a huge number of papers explaing that pre-play interaction has positive effects on cooperation, here the pre-play has not, I need a discussion.

9. It is said that when moving from drinking in groups of 4 to playing the public good game they did not stay in the same group. Why ?

10. I do not see any results about the differences between those who had drinks alone and in a group.

11. It is sais they made 4 decisions, that means there were 4 rounds in the public good game?

12. I see the experiment as very invasive with all the measures etc...so this would compensate on the social effect of drinking, right? The situation is not "normal" and this is not how cooperation decisions are taken "in real life" after a drink. So this can explain the results as participants understand that they are expected to be inhebriated and their actions under this condition will be studied. So this can induce behavior. I would need some discussion.

6. PLOS authors have the option to publish the peer review history of their article (what does this mean?). If published, this will include your full peer review and any attached files.

Reviewer #1: No

Reviewer #2: **Yes: **Angela Sutan

---

## [Decision Letter · Decision Letter 1]

16 Apr 2021

PONE-D-21-04756R1

Alcohol Unleashes Homo Economicus by Inhibiting Cooperation

PLOS ONE

Dear Dr. Zak,

Thank you for submitting your manuscript to PLOS ONE. After careful consideration, we feel that it has merit but does not fully meet PLOS ONE’s publication criteria as it currently stands. Therefore, we invite you to submit a revised version of the manuscript that addresses the points raised during the review process.

I will explain my decision in detail.

1. Let me start saying that one referee is still against the paper (see the report below) while the other withdraw (he/she did not mention why).

2. Typically I follow the advice of the referees as they are but this time I need to do my own report since Reviewer #1 is gone.

My own impression.

1. I agree with Reviewer #2 that you need more data. But I also agree with you that given the current circumstances is quite hard to run these type of experiments.  

2. I also agree with Reviewer #2  (and #1) that you have too many treatments (not 2 but 4) and obviously you may have a problem with power. 

3. I also agree with Reviewer #2  that the idea of replacing teams members with RAs (intoxicated or not) is a least exotic.

4. Lastly, still I do not understand if subjects played with or without feedback (and this is NOT trivial at all).

Ok. Personally I find the paper interesting. Yes it is. But I feel that we certain things need to be fixed in order to be sure that the results that you get are robust.

There are 3 complementary ways to solve some of the problems (regarding the PGG).

W1: Instead of running the analysis shown in the paper, just need to do is to use a panel. You use for each subject the four decisions, the treatment, the second treatment, etc... and the feedback in t-1 (if does exist). If you have feedback then you can use only t=2, 3 and 4. Typically you can run the same model with and without feedback. And you can use alcohol as continuous or dummy. ... Your main claims (contributions and earnings) would be much clear using panel. Besides my own taste, panel is the standard econometric procedure in this literature.

W2 (robustness): In order to remove any potential problem of the introduction of RAs as participants you can run a regression for t=1. In this case, does not matter whether the RA contributed the median, high or low since participants only play against their own beliefs. 

W3 (robustness 2): doing the same for the last round where you add as variables the feedback (if exist) for round 1, 2 and 3. This is important because the RA would appear just once in each group (or none). 

There are many other comments of Referee #2 that I agree but I also understand that every single person has a different way to explain things. 

Sorry for being so repetitive but I tried to my best. As author I much prefer to get constructive and doable feedback.

Best, Pablo

** the rest of the message is automatic ***

We look forward to receiving your revised manuscript.

Kind regards,

Pablo Brañas-Garza, PhD Economics

Academic Editor

PLOS ONE

Reviewers' comments:

Reviewer's Responses to Questions

**Comments to the Author**

1. If the authors have adequately addressed your comments raised in a previous round of review and you feel that this manuscript is now acceptable for publication, you may indicate that here to bypass the “Comments to the Author” section, enter your conflict of interest statement in the “Confidential to Editor” section, and submit your "Accept" recommendation.

Reviewer #2: (No Response)

2. Is the manuscript technically sound, and do the data support the conclusions?

Reviewer #2: Partly

3. Has the statistical analysis been performed appropriately and rigorously? 

Reviewer #2: I Don't Know

4. Have the authors made all data underlying the findings in their manuscript fully available?

Reviewer #2: Yes

5. Is the manuscript presented in an intelligible fashion and written in standard English?

Reviewer #2: Yes

6. Review Comments to the Author

Reviewer #2: Dear authors, with regard to the revisions performed in this round, I have the following comments listed here together with the previous questions and replies.

1. The sample is clearly too small. 128 individuals, split into 2 treatments (control and intervention), plus in 2 again (drink alone or drink with 3 people). I would expect more sessions to be conducted. A power of test calculation is now added to the revised ms showing the majority of statistical tests have high power. Most of the analysis uses the full sample to keep the power of tests high, identifying subconditions such as drinking alone with binary indicator variables. With the current pandemic and on advice from the editor, we are unable to conduct additional sessions

of this experiment. We have added a cautionary note in the Discussion that segmenting the sample will reduce the power of the statistical tests.

COMMENT: We all know that the paper will be cited without reference to the footnote or the low power of tests. People will only cite the results, not their limited power (especially as the author is famous). I cannot understand why sessions could not be conducted later, after the pandemics. Why the hurry?

2. I would need more details about the sessions. It is said that a session consists of 12 people, how

were exactly composed the sessions since with 128 people you cannot have only sessions of 12?

The paper states that "Typical sessions included 12 participants." Some sessions were smaller

or larger due to dropouts and a research assistant filled in the group of 4 with the fill-in data

not analyzed.

COMMENT: The authors have now written in the paper : "Sessions that had less than 12 participants used a research assistant to complete the group of four. The research assistant always made the median decision

from the existing literature and these data were not included in the analysis. Participants were randomly assigned to groups of four in order to reduce any pre-play social attachment. Participants were not informed about the PGG while drinking in order to reduce anticipatory effects."

This problem is huge. first of all, a typical session should be a systematic session. Given the low number of people (12) and the random matching, all sessions need to be conducted with 12 participants exaclty. Otherwise the probabilities of rematching are not the same. Second, the participants are not told that an RA will play and how. They think they play agains a regular participant. Third, what is the "median from the literature"? This has clearly an influence because participants randomly meet someone playing a median, and possibly not a free rider or so. Specially that this paper claims not to confirm median results from the literature. Also, OK those numbers are said to be removed from the data but: there is random matching, data is not independent, all the session has to be dropped. This is clearly what we explain to all students starting doing experiments. Later on, they will read this paper written by a famous author in which it is allowed in a random matching game to introduce without telling the participants a player playing a median from the literature...Finally, before playing the PGG, some participants were interacting in groups of 4, they were drinking together. So they were not 4 in some groups, or was the RA also drinking with them? Or they were 3 and after were told that they are 4 in the PGG?

3. Why were there mixed groups? There is a large litterature showing that men and women react differently to alcohol and behavioural tasks, why not making groups of females only and of men only?

The revised ms now clarifies that we sought to capture what typically happens when people go

to a bar.

COMMENT: I cannot understand why the authors insisted on creating the atmosphere from the bar as later (see the paragraph commented at the previous point) they say this anyway was supposed to be destroyed by the random matching and also later in another response the authors say that the paper was not supposed to create real consumption conditions (when I asked abou the invasive protocol). I cannot understand now with the new comments why the authors needed this variation of the treatments in groups of 4 while drinking.

4. It is said there was no deception, however some participants had placebo and it is said they were pushed to believe they will have alcohol. So I am sorry, but I cannot understand how this was not deception?

Participants were informed they would consume alcohol or placebo, hence no deception.

COMMENT: This is not said in the text. In the text it is written (unless I am wrong) that there was a placebo and alcohol, and that participants were made to think it would be alcohol. It should be written clearly in the text : participants were informed there was maybe placebo. I had no access to instructions anyway.

5. It is not clear if participants knew the task in advance, while drinking, and if they knew there will be a task at all?

The revised ms. clarifies that during the consumption phase, participants did not know what tasks they would do later.

OK. But they knew they will be doing something later, after the drinking, right?

6. It is explained that alcohol makes people act on impulse, but economic papers show that when acting on impulse, people are more cooperative. So I would appreciate a discussion. The revised ms. Discussion now brings in more literature on impulsivity and economic decisions. Thank you for this suggestion.

OK.

7. It is said that alcohol motivates the desire for immediate reward, but this reward could be warm glow or something like this, so, again, I would need a discussion. We have added a cite to the Discussion on this.

OK.

8. There is a huge number of papers explaing that pre-play interaction has positive effects on cooperation, here the pre-play has not, I need a discussion. Great suggestion, we have added this to the Discussion. The role of pre-play was reduced by randomly matching participants in groups of four for the PGG.

OK.

9. It is said that when moving from drinking in groups of 4 to playing the public good game they did not stay in the same group. Why ? Revised ms. clarifies in Methods that the PGG used random matching of participants --the goal

was to reduce the effect of pre-play interactions.

COMMENT: So, again, in this case why bother with preplay in groups of 4? I suppose I miss something, and now even more than after the first version.

10. I do not see any results about the differences between those who had drinks alone and in a group. We now mention this nonfinding on PPGg and PPGc to the first paragraph of the Results.

OK.

11. It is sais they made 4 decisions, that means there were 4 rounds in the public good game?

Correct, with random rematching.

OK.

12. I see the experiment as very invasive with all the measures etc...so this would compensate on

the social effect of drinking, right? The situation is not "normal" and this is not how cooperation

decisions are taken "in real life" after a drink. So this can explain the results as participants

understand that they are expected to be inhebriated and their actions under this condition will be

studied. So this can induce behavior. I would need some discussion.

The study did not claim to replicate real life. Rather, we sought to establish the role of a

commonly used drug (alcohol) on cooperation. My lab has published many studies using blood

draws to establish neurochemical mechanisms affecting behavior and choice data are very

similar to that found without blood draws. We now mention this in the Discussion with a

citation.

COMMENT: I am aware of all paper published in your lab. With regard to this, I know that this paper will be read and cited ans used as an example by young scholars (or cited in the press). So my previous comments insist on paying attention to the way in which this paper was conducted (problems in the procedures, in the design, low power, need to drop incomplete sessions, conduct others). Yet, with regard to this specific remark (number 12): here I comment only on this paper, not on the other papers from your lab. It is a good thing to measure blood indicators and everything. I say here in this context of drinking, this may be too invasive. But now, anyway, I cannot understand what you wanted to do, because in some places you say that a bar consumption was intended to be replicated, and in some other places that pre-play interactions was supposed to be counterbalanced by random matching and that the paper was not intended to replicate real life. So in that case, if the study intended to look at the effect of alcohol only, I see no use of sessions of 4, and also, I come back to my remark: pre-play in miwed groups will induce expectations about cooperation from women, domination feelings etc....and this has to be controlled.

I want to say that study like this one are important. And that this paper is likely to become a reference. But will all respect, it needs to respect the standards as to be clean.

7. PLOS authors have the option to publish the peer review history of their article (what does this mean?). If published, this will include your full peer review and any attached files.

Reviewer #2: No

---

## [Author Response · Author response to Decision Letter 1]

30 May 2021

Letter uploaded with files, but pasted here again.

We thank R2 for taking the time to offer additional suggestions to improve our ms. We looked at each one in detail and have revised the ms accordingly. My coauthors and I believe this revision addresses the suggestions you have made and herein we detail the changes. We expect you will find this revision satisfies your remaining questions. 

Power. As we note in the cover letter, the bulk of our analyses use the full N=124 data set with indicators for treatment and socializing so these analyses are not underpowered. In addition, as suggested by the editor, we have done a panel data analysis with the full set of n=496 behavioral observations (rather than averaging PGG behavior). The revised ms. now reports the panel data analysis showing that alcohol continues to reduce PGG contributions. The editor also recommended analyzing alcohol's effects on round 1 and round 4 contributions. Again, the results for alcohol reducing PGG contributions are sustained as the paper now reports. We appreciate these excellent suggestions that show the robustness of the findings. 

For these reasons, we disagree that more data is needed. The only underpowered tests reported are for free riders and the effect of drinking alone, both secondary analyses that the paper includes but are not core findings. Further, both these findings are statistically significant even with lower power. In addition, the drinking alone condition was only significant in affecting the stress response and was not otherwise used in the analysis so that sample is not split in half due to this. Indeed, we believe the hypothesis of whether it is alcohol or socializing that affects behavior was worth testing and reporting even if it did not affect behavior. 

Replacements. There were only 4 instances of RAs filling in and these data were not analyzed as the paper states. The Methods of the revised ms now expands on this and clarifies that RAs did not drink and could not be identified by participants as decisions were anonymous with the RA filling in a PGG group entering data from the control room. Since participants never received feedback on others' contributions, the revised paper emphasizes there is no way for this filling to affect participants' decisions.

Mixed sexes and design approach. This was a design choice to capture the most common effects for an already complicated experiment (e.g. we hired a phlebotomist to do 248 blood draws, then we processed these tubes to isolate plasma that was stored in an ultracold freezer until sent for analysis, mixed drinks, measured BAC multiple times, etc.). My lab is focused on identifying mechanisms producing behavior which is why we took blood samples before and after drinking. Lab studies balance what people do outside the lab (e.g. drink) with the precision that lab studies offer in measuring behavior and the mechanisms that produce them. There are many ways to study the effects of drinking on cooperation, but our finding of stress and negative affect are unique to the literature and help explain why alcohol reduces cooperation. We have added to the Discussion that our result may not generalize to single sex groups or older individuals as you noted. Our hope is that this study provokes others to explore different ways to study this important topic. 

Deception: The revised Methods now clarifies that the instructions to participants stated that only one-half would drink alcohol while the others would receive a placebo. The masking of the placebo followed a protocol in the cited alcohol literature as we now emphasize. 

Tasks. The revised ms. clarifies that participants only knew they would "make decisions," but not what decisions they would make.

---

## [Editor Report · Decision Letter 2]

2 Jun 2021

Alcohol Unleashes Homo Economicus by Inhibiting Cooperation

PONE-D-21-04756R2

Dear Dr. Zak,

We’re pleased to inform you that your manuscript has been judged scientifically suitable for publication and will be formally accepted for publication once it meets all outstanding technical requirements.

Kind regards,

Pablo Brañas-Garza, PhD Economics

Academic Editor

PLOS ONE
---

## [Editor Report · Acceptance letter]

14 Jun 2021

PONE-D-21-04756R2 

Alcohol unleashes homo economicus by inhibiting cooperation 

Dear Dr. Zak:

I'm pleased to inform you that your manuscript has been deemed suitable for publication in PLOS ONE. Congratulations! Your manuscript is now with our production department. 

Kind regards, 

on behalf of

Dr Pablo Brañas-Garza 

Academic Editor

PLOS ONE